# Proteoglycan from *Bacillus* sp. BS11 Inhibits the Inflammatory Response by Suppressing the MAPK and NF-κB Pathways in Lipopolysaccharide-Induced RAW264.7 Macrophages

**DOI:** 10.3390/md18120585

**Published:** 2020-11-24

**Authors:** Qingchi Wang, Weixiang Liu, Yang Yue, Chaomin Sun, Quanbin Zhang

**Affiliations:** 1CAS Key Laboratory of Experimental Marine Biology, Institute of Oceanology, Chinese Academy of Sciences, Qingdao 266071, China; wangqingchi@qdio.ac.cn (Q.W.); liuweixiang@qdio.ac.cn (W.L.); yueyang@qdio.ac.cn (Y.Y.); sunchaomin@qdio.ac.cn (C.S.); 2Laboratory for Marine Biology and Biotechnology, Qingdao National Laboratory for Marine Science and Technology, Qingdao 266071, China; 3Department of Earth Science, University of Chinese Academy of Sciences, Beijing 100049, China; 4Center for Ocean Mega-Science, Chinese Academy of Sciences, Qingdao 266071, China

**Keywords:** *Bacillus* sp., proteoglycan, anti-inflammation, macrophages

## Abstract

Inflammation is involved in the pathogenesis of many debilitating diseases. Proteoglycan isolated from marine *Bacillus* sp. BS11 (EPS11) was shown to have anticancer activity, but its anti-inflammatory potential remains elusive. In the present study, the anti-inflammatory effects and mechanism of EPS11 were evaluated using a lipopolysaccharide (LPS)-induced RAW264.7 macrophage model. Biochemical characterization showed that the total sugar content and protein content of EPS11 were 49.5% and 30.2% respectively. EPS11 was composed of mannose, glucosamine, galactosamine, glucose, galactose, rhamnose, and glucuronic acid. Its molecular weight was determined to be 3.06 × 10^5^ Da. The protein determination of EPS11 was also performed. EPS11 displayed a strong anti-inflammatory effect on LPS-stimulated RAW264.7 macrophages in vitro, which significantly suppressed inflammatory cytokines and mediators (such as NO, TNF-α, IL-6 and IL-1β, and COX-2). Western blot analysis indicated that EPS11 could downregulate the expression of many key proteins in mitogen-activated protein kinases (MAPKs) and transcription factor nuclear factor-κB (NF-κB) signaling pathways. In particular, EPS11 almost completely inhibited the expression of NF-κB P65, which indicated that EPS11 acted primarily on the NF-κB pathways. These findings offer new insights into the molecular mechanism underlying the anti-inflammatory effect of EPS11.

## 1. Introduction

The immune system has multiple functions, including recognition, elimination, and regulation, which are beneficial for organisms. However, the dysregulation of the immune system may cause many pathological immune responses, such as inflammation, allergy, and asthma [1]. Inflammation has been especially linked with the pathogenesis of many diseases, such as arthritis, cancer, neurodegenerative disease, cardiovascular disease, and multiple sclerosis [2,3,4,5,6,7]. Therefore, a number of chemically synthesized compounds were used as immunomodulators to regulate the immune system [8,9]. Due to some adverse events, such as toxic effects on the liver, lungs, kidneys, and gastrointestinal tract, their clinical applications are restricted [10]. In recent years, much attention has been paid to natural polysaccharides due to their nontoxic properties and immunomodulatory activities [11,12,13].

Macrophages act as the first line of defense in the immune system, and display extensive biological functions such as phagocytosis, the destruction of pathogens, antigen presentation, and cytokine production [14]. Many external stimuli, such as lipopolysaccharides (LPS), carbohydrates, and some cytokines can interact with the pattern-recognition receptors of macrophages. Then, a series of intracellular responses are activated, and several related signaling pathways, including transcription factors nuclear factor-κB (NF-κB) and mitogen-activated protein kinases (MAPKs) may be involved in signal transmission [15,16]. During inflammation, macrophages execute their function of immune response by producing various inflammatory cytokines and mediators [17,18], such as tumor necrosis factor (TNF-α), transforming growth factor-β (TGF-β), interleukin-1 (IL-1), interleukin-6 (IL-6), prostaglandin E2 (PGE2), and nitric oxide (NO) [19,20]. In turn, these inflammatory mediators promote the development of inflammation. Thus, compounds that can reduce inflammatory mediators may be developed into therapeutic agents for inflammatory diseases.

Proteoglycan (EPS11) was sourced from marine *Bacillus* sp. BS11, which was reported to inhibit cancer-cell growth via blocking cell adhesion in previous studies [21,22]. However, the inflammatory effects and mechanism of EPS11 were not investigated. In this study, EPS11 was prepared, and its structural features were determined by chemical analysis. The in vitro immunomodulatory effect of EPS11 was investigated by determining the inhibition of inflammatory mediators produced by LPS-activated RAW 264.7 macrophages. The mechanisms of EPS11 acting on MAPKs and NF-κB signaling pathways were also explored.

## 2. Results

### 2.1. Extraction and Purification of EPS11

The extraction and purification of EPS11 was carried out as described in Section 4.2. The polysaccharide fraction and protein fraction were detected by sulfuric acid–phenol method and UV detector, respectively (Figure 1). The results showed that the eluted peak was relatively high and symmetrical when NaCl concentration was 250 mmol/L (Figure 1A,C). Then, this peak was collected, concentrated, and purified by HiLoad^TM^ 16/600 Superdex^TM^ 200 gel-column chromatography (Figure 1B,D). The highest peak in gel chromatography was collected to concentrate, dialyze, and lyophilize, and it was labeled as EPS11.

### 2.2. Physicochemical-Property Analysis of EPS11

Table 1 shows that total sugar content is 49.5%, which was determined using Man as the standard. The protein content was 30.2% and the moisture content was 5.6%. The results showed that EPS11 is a proteoglycan. The results of amino acid analysis showed that EPS11 contained 20 kinds of amino acids, and the total amino acid content was 30.5%, which was consistent with the results of protein content (Table 2).

The chromatograms of monosaccharide standards and EPS11-sourced hydrolysate are presented in Figure 2A,B. EPS11 was composed of mannose (Man), glucosamine (GlcN), galactosamine (GalN), glucose (Glc), galactose (Gal), rhamnose (Rha), and glucuronic acid (GlcA) with detected molar ratio as 13.08:7.84:8.23:4.73:7.03:1.00:1.03 (Table 1). The results showed that Man was the dominant monosaccharide in EPS11, and GlcN, GalN, Glc, and Gal were also present in EPS11 with relatively high abundance. As shown in Figure 2C, the peak of EPS11 at 13.762 min was single and symmetrical, and average molecular weight was determined for 3.06 × 10^5^ Da (Table 1). 

The FT–IR spectrum of EPS11 presented in Figure 2D showed the characteristic absorption. The broad intense peak at 3272 cm^−1^ and a weak peak around 2948 cm^−1^ represented the stretching vibration of O–H or N-H bonds and C–H bonds in the sugar ring, respectively. Strong peaks at 1634 and 1540 cm^−1^ were associated with the stretching vibration of C=O or formation vibration of N–H. This indicated that aminosaccharides or protein were present. The narrow intense peak at 1051 cm^−1^ represented the stretching vibration of C–O–C or C–O–H. Peaks at 1225 and 836 cm^−1^ implied that the sulfate group might have been present.

### 2.3. Protein Profiling of EPS11 by LC-MS/MS

To investigate the composition of protein contained in EPS11, LC-MS/MS analysis was carried out. The results showed that a total of 41 peptides were obtained by enzymatic hydrolysis from EPS11. The sequences of the peptides are shown as follow: AFSEDGGTDIDLLEAGEWIIAPK; AFSEDGGTDIDLLEAGEWIIAPKDAEGNPHPEK; AGENVGVLLR; AIDKPFLLPIEDVFSISGR; AIQEPNCLEATIAPSGHK; ALVTGGLFR; APDVVNNSWGGGPGLDEWYR; APDVVNNSWGGGPGLDEWYRPMVQNWR; APEEEGNYMIR; DAEGNPHPEK; DAEGNPHPEKAPDVVNNSWGGGPGLDEWYR; DAEGNPHPEKAPDVVNNSWGGGPGLDEWYRPMVQNWR; DAEGNPHPELAPDVVNNSWGGGPGLDEWYRPMVQNWR; DSFGNETRK; DSYVGDEAQSK; EITALAPATMK; FWNTEWPNPGGTNFK; GDVHDENLAWVK; GYRPQFYFR;HQGVMVGMGQK; HTPFFK;HYAHVDCPGHADYVK; IWHHTFYNELR; KTAEGLLNINTK; LCYVALDFEQEMATAASSSSLEK; MSLAEGQER; PDVSAPGVNIR; PMVQNWR; QEYDESGPSIVHR; QLELLEKMRDMNASLSK; SYELPDGQVITIGNER; TTLTAAITTVLAK; VAPEEHPVLLTEAPLNPK; VEIHDASGPDGAPGK; VGAPEVWDMGIDGAGTVIANIDTGVQWDHPALMEQYR; VKDSFGNETR; VKDSFGNETRK; VLTGGVDANALQRPK; YHEEIPLR; YTLAGTEVSALLGR. After mass spectrometry data retrieval and screening, the potential proteins in EPS11 showed some homology with the identified eight proteins, which are shown in the Table 3. The two proteins with the highest matching scores were Bacillopeptidase F and Actin. Bacillopeptidase F is a serine endopeptidase, which was first isolated from Bacillus subtilis. Actin is commonly found in eukaryotic cells and is encoded by highly conserved genes.

### 2.4. Effects of EPS11 on RAW264.7 Macrophage Viability

To evaluate the effect of EPS11 on the viability of RAW264.7 macrophages, the macrophages were treated with EPS11 (6.25–200 μg/mL) or LPS (1.0 μg/mL) for 24 h. The control group was designated as 100% cell viability. The results showed that EPS11 had no cytotoxicity with RAW264.7 macrophages at the tested concentrations (6.25–200 μg/mL; Figure 3A). Figure 3B displays that RAW264.7 macrophages were coincubated with EPS11 and LPS, and cell viability exceeded 75% in all groups, indicating that EPS11 was nontoxic to RAW264.7 macrophages. Therefore, these concentrations (6.25–200 μg/mL) were used for the following experiments.

### 2.5. Active Fraction Appraisal

To identify the active fraction, LPS-induced RAW264.7 macrophages were treated with EPS11 (200 μg/mL) which was treated by proteinase K and NaIO4, respectively. The results showed that there was no significant effect on the inhibition of NO production with EPS11 treated by proteinase K. However, the cells incubated with NaIO4-treated EPS11 increased NO production as the same as LPS group (Figure 4). The results suggested that the anti-inflammatory activity of EPS11 is due to the polysaccharide fraction.

### 2.6. EPS11 Inhibits NO Production

NO, as an important indicator of inflammation, is produced and regulated by nitric oxide synthase (iNOS) [23]. As shown in Figure 5A, EPS11 inhibited the generation of NO in LPS-induced RAW264.7 macrophages in a concentration-dependent manner. In particular, macrophages treated with EPS11 (50–200 μg/mL) showed significantly lower NO production than that in the control. NO inhibitory effects of positive controls aspirin (200 μg/mL) and NG-monomethyl-l-arginine, monoacetate salt (l-NMMA) (50 μmol/L) were also observed. Figure 5B shows that EPS11 with different concentrations did not influence the generation of NO in RAW264.7 macrophages without LPS stimulation, which indicated that EPS11 may act by influencing NO production-related signaling pathways.

### 2.7. EPS11 Decreases Expression of COX-2, TNF-α, IL-1β, and IL-6

The inhibitory effects of EPS11 on COX-2, TNF-α, IL-1β, and IL-6 levels were further investigated. Figure 6 shows that LPS (1.0 μg/mL) significantly increased the expression of COX-2 and cytokines in macrophages. However, the production of TNF-α in supernatants of the EPS11-treated macrophages was significantly decreased in a concentration-dependent manner (Figure 6A). Similarly, the inhibitory effects of EPS11 on IL-1β, IL-6, and COX-2 in LPS-induced RAW264.7 macrophages were also observed (Figure 6B–D). Therefore, these observations suggested that EPS11 may inhibit the protein expression of COX-2, TNF-α, IL-1β, and IL-6. 

### 2.8. Effects of EPS11 on MAPK Pathways

To explain the mechanism of the anti-inflammatory effects of EPS11 on macrophages, Western blot analysis was applied to assess the effects of EPS11 on MAPK signaling pathways. According to the immunoblots in Figure 7A, phosphorylated P38, JNK, and ERK1/2 were significantly increased in the LPS group. In the test concentrations, EPS11 dramatically depressed the phosphorylated P38, JNK, and ERK1/2, and nonphosphorylated ERK1/2 in a concentration-dependent manner (Figure 7B–D). However, nonphosphorylated P38 and JNK were unchanged in all groups. This indicated that EPS11 may act by inhibiting the phosphorylation of P38, ERK1/2, and JNK, and the expression of ERK1/2 in MAPK pathways.

### 2.9. Effects of EPS11 on NF-κB Pathway

IκB and P65 protein levels were determined to unravel the anti-inflammatory mechanism of EPS11 on the NF-κB pathway in RAW264.7 macrophages. As shown in Figure 8A, EPS11 (200 μg/mL) could significantly suppress phosphorylated IκB in LPS-stimulated macrophages, while other concentrations (50 and 100 μg/mL) of EPS11 had no remarkable effect (Figure 8B). Various concentrations of EPS11 (50, 100, and 200 μg/mL) significantly decreased phosphorylated P65 in a concentration-dependent manner, and 200 μg/mL EPS11 completely restrained the phosphorylation of P65 (Figure 8C). The nonphosphorylation of P65 was also almost fully inhibited with EPS11 in the range of 50-100 μg/mL (Figure 8D). This indicated that EPS11 may inhibit the expression or enhance the degradation of NF-κB P65. Therefore, results revealed that EPS11 suppressed LPS-stimulated NF-κB activation by mainly preventing the phosphorylation or expression of P65.

## 3. Discussion

In the current study, the inhibitory effects and mechanism of EPS11 against LPS-stimulated inflammatory response were investigated. The results revealed that EPS11 isolated from *Bacillus* sp. BS11 exhibits anti-inflammatory properties, which appeared to be attributable to the inhibition of the activation on MAPK and NF-κB signal pathways in LPS-induced RAW 264.7 macrophages.

It is well known that inhibiting the production of inflammatory cytokines or regulating the balance of key proteins is a possible strategy for treating inflammation [24,25]. Therefore, the generation of NO, IL-1β, IL-6, TNF-α, and COX-2 was determined in LPS-stimulated RAW264.7 macrophages treated with EPS11 (Figure 5 and Figure 6). Data showed that EPS11 can effectively downregulate those inflammatory mediators. The results were consistent with those in a report by Kumar et al., who found that an exopolysaccharide isolated from *Kocuria rosea* strain BS-1 suppressed the release of reactive oxygen species (ROS), NO, TNF-α, and IL-6 from LPS-induced RAW 264.7 macrophages [26]. NO is produced and regulated by Inducible Nitric Oxide Synthase (iNOS), which is activated by immunostimulatory cytokines (such as IL-1β, IL-6, and TNF-α) through the activation of inducible nuclear factors, including NF-κB [23]. The decrease in inflammatory mediators indicated that the activity of inflammation-related signaling pathways was inhibited.

To explore the mechanism of anti-inflammation by EPS11, MAPK signaling pathways were first investigated. MAPK pathways, which mainly included the pathways of P38, ERK1/2, and JNK, could be stimulated by stimuli such as LPS, IL-6, and TNF-α to trigger the inflammation responses. These stimuli could activate ERK, JNK, and P38, and then act on their respective substrates to affect the activity of a variety of transcription factors, thereby regulating the gene expression of various cytokines such as TNF, IL-1, IL-6, and IL-8. In turn, TNF-α, IL-1, and other inflammatory mediators could activate different MAPKs and regulate the production of other inflammatory mediators by promoting or inhibiting the transcription of genes [27,28]. Some natural polysaccharides were reported to inhibit the activation of MAPKs. Wu et al. reported that a sulfated polysaccharide extracted from *Sargassum cristaefolium* could effectively reduce the phosphorylation of P38, ERK, and JNK in LPS-stimulated RAW264.7 macrophages, so as to suppress the expression of iNOS [29]. In this study, EPS11 was found to decrease the phosphorylation of P38, ERK1/2, and JNK in a similar way (Figure 7). This indicated that EPS11 could suppress the activation of MAPK pathways and reduce the intracellular transduction of inflammatory signals, which could downregulate the production of inflammatory mediators by inhibiting the transcription of genes. This might represent a possible mechanism underlying the anti-inflammatory activity of EPS11.

As the downstream-signaling molecules of MAPKs, NF-κB plays a vital role in the progression of inflammation [30]. NF-κB is a dimer protein containing two subunits (P65 and P50) that are tightly inhibited by forming a stable IκB–NF-κB complex in macrophages [31]. IκBα is degraded by the IκB kinase complex (IKK) when macrophages are stimulated by the upstream signal, which allows for the NF-κB proteasome to activate and rapidly translocate into the nucleus [32,33]. In the nucleus, the two subunits (especially P65) bind to a homologous DNA site to start the transcription of relevant genes, which potentially promotes the occurrence and progression of inflammation [34]. The activation of NF-κB in macrophages can promote the expression of proinflammatory cytokines (such as TNF-α, IL-1, and IL-6) and differentiation into the M1 macrophage phenotype [35]. In the present study, EPS11 could significantly block the LPS-stimulated phosphorylation of IκB and P65 (Figure 8). It was proven that inhibiting the phosphorylation of P65 can selectively suppress the expression of many genes associated with inflammatory factors [36]. This suggested that EPS11 not only inhibits the degradation of IκB from the IκB–NF-κB complex, but also prevents the binding of NF-κB to its DNA binding site. Interestingly, EPS11 also inhibited nonphosphorylated P65 in EPS11-treated macrophages. In addition to phosphorylation, another regulatory mechanism (ubiquitination) is activated when NF-κB transports to nucleus. NF-κB is ubiquitinated by the E3 ubiquitin ligase complex and ultimately leads to the degradation of the NF-κB heterodimer protein [37,38]. Ubiquitination is regulated by deubiquitinase ubiquitin specific protease-7 (USP7), which specifically requires P65 as the substrate. The inactivation of USP7 activity results in the reduced expression of target genes and P65 degradation [39]. This implied that EPS11 may suppress the expression of P65 by enhancing its ubiquitination and degradation. Therefore, it is possible that EPS11 inhibited the phosphorylation and promoted the ubiquitination of NF-κB P65 to decrease the transcription and expression of inflammatory mediators and prevent macrophages from differentiating into the M1 phenotype.

## 4. Experiments and Methods

### 4.1. Materials and Reagents

*Bacillus* sp. BS11 was isolated from marine mud samples collected near the Yap Trench (1143 m deep) in the tropical Western Pacific (139°3802′ E, 11°44162′ N); the RAW 264.7 macrophage line was purchased from Macrophage Resource Center, Shanghai Institute of Life Sciences, Chinese Academy of Sciences (Shanghai, China); 1-phenyl-3-methyl-5-pyrazolone (PMP) was purchased from Sigma-Aldrich (St. Louis, MO, USA); Dulbecco’s Modified Eagle’s Medium (DMEM)/high-glucose medium was purchased from HyClone (Logan, UT, USA); lipopolysaccharides (LPS) and aspirin were purchased from Yuanye (Shanghai, China); 3-(4, 5-dimethylthiazol-2-yl)-2, 5-diphenyltetrazolium bromide (MTT) reagent and dimethyl sulfoxide (DMSO) were purchased from Sigma-Aldrich (St. Louis, MO, USA); fetal bovine serum (FBS) was purchased from Gibco Life Technologies (Grand Island, NY, USA); ELISA kit for the analysis of cyclooxygenase (COX-2), TNF-α, IL-1β, and IL-6 production was obtained from AndHider (Qingdao, Shandong, China); Griess reagent, NG-monomethyl-l-arginine, monoacetate salt (l-NMMA), and horseradish peroxidase (HRP)-conjugated goat antimouse IgG secondary antibodies were purchased from Beyotime (Nantong, Jiangsu, China); antibodies to ERK, phosphorylated ERK (P-ERK), JNK, phosphorylated JNK (p-JNK), P38, phosphorylated P38 (P-P38), IκBα, phosphorylated IκBα (p-IκBα), P65, phosphorylated P65 (P-P65), and β-actin were purchased from Santa Cruz (Dallas, TX, USA); antibodies to GADPH and α-tubulin were purchased from Abcam (Cambridge, MA, USA); HRP-conjugated goat antirabbit IgG secondary antibodies were purchased from Affinity (Chicago, IL, USA).

### 4.2. Extraction and Purification of EPS11 from Bacillus sp.BS11

The pure culture of marine *Bacillus* sp.BS11 was inoculated to a 2216E medium (0.5% tryptone, 0.1% yeast extract powder, and 1% sucrose; pH = 7.5) in a proportion of 0.1% (*v*/*v*) and cultured at 28 °C and 140 RPM for 24 h. To obtain the precipitation of the crude polysaccharide, the supernatant, which was obtained by centrifugation (8000 RPM, 20 min) of the fermentation liquid, was precipitated with triple alcohol by volume. Precipitation was again dispersed in pure water, and mixed with Sevag’s reagent (chloroform:N-butyl alcohol = 5:1, *v*/*v*) in a ratio of 4:1 to remove the proteins [40]. The crude polysaccharide was obtained with the dialysis (MWCO: 8.0 kD), concentration, and lyophilization of the supernatant.

To obtain a homogeneous polysaccharide, diethylaminoethyl cellulose (DEAE) Fast Flow anion-exchange column was devoted to purifying the crude polysaccharide using 50 mmol/L NaCl with 20 mmol/L Tris-HCl buffer (pH = 9.0) and a linear gradient eluent (from 50 to 500 mmol/L NaCl solution with 20 mmol/L Tris-HCl buffer). The fractions were recorded and collected by AKTA Purifier (GE Healthcare, USA). The collected polysaccharide fractions continued to be purified with a HiLoad^TM^ 16/600 Superdex^TM^ 200 gel chromatographic column (GE Healthcare, USA), which used 150 mmol/L NaCl and 20 mmol/L Tris-HCl buffer (pH = 9.0) as the mobile phase. The final polysaccharide fraction (EPS11) was collected, dialyzed, and lyophilized.

### 4.3. Determination of Physical and Chemical Properties

Total EPS11 content was determined by the sulfuric acid-phenol method, which uses Man as standard to draw the standard curve. The protein content was determined using a bicinchoninic acid (BCA) kit (Solarbio, Beijing, China). Moisture content was measured by the ambient-pressure-drying method, which needs the sample to be dried to a constant weight. The amino acid composition of EPS11 was determined using an automatic amino acid analyzer (L-8500A, Hitachi, Japan).

The monosaccharide composition of EPS11 was measured by the method of precolumn derivation RP-HPLC [41,42]. EPS11 (5.0 mg/mL) was hydrolyzed in a sealed tube with conditions of 2.0 mol/L trifluoroacetic acid (TFA), 105 °C, and 4 h. The TFA was neutralized by NaOH solution. Then, the hydrolysate or monosaccharide standard was derivatized with 0.5 mol/L PMP-methanol (120 μL) and 0.3 mol/L NaOH (100 μL) for 1 h at 70 °C. The reaction liquid was extracted by chloroform three times after it was neutralized with HCl. The derivatives were filtered with a micron membrane filter (0.22 μm) to perform HPLC analysis under the given chromatographic conditions (Table 4). The molar ratio of the monosaccharides was calculated on the basis of the peak area and relative molecular mass of each monosaccharide.

High-performance gel-permeation chromatography (HPGPC) was carried out to determine the molecular weight of EPS11 with a refractive-index detector. EPS11 solution (5 mg/mL) was eluted by 0.1 mol/L Na_2_SO_4_ through a SHODEX SUGAR KS-804 (7.8 × 300 nm) column (SHODEX, Tokyo, Japan). Column temperature and flow rate were set as 35 °C and 0.5 mL/min, respectively. The calculation of the molecular weight was according to the standard curve that was plotted with the retention time and the molecular weight of dextran (5250, 9750, 13,150, 36,800, 64,650, 135,350, and 300,600 Da).

### 4.4. Infrared Spectrum

The FT–IR spectrum was recorded on a Nicolet iS 10 FT–IR spectrometer (Thermo Fisher, Waltham, MA, USA) in the range of 400–4000 cm^−1^.

### 4.5. Protein Identification by LC-MS/MS

EPS11 solid was added to a trypsin buffer (6µg Trypsin in 40μL NH_4_HCO_3_ buffer), and the mixtures were incubated at 37 °C for18 h. The peptides were desalted using C18 StageTip (Thermo Scientific) and separated using an EASY-nLC 1200 Nano UHPLC system (Thermo Fisher, Waltham, MA, USA) equipped with a Trap Column (100 µm 20 mm, 5 µm, C18, Dr. Maisch GmbH, Ammerbuch-Entringen, Germany). The sample was injected and separated by a chromatographic column (75 µm 150 mm, 3 µm, C18, Dr. Maisch GmbH) at a flow rate of 300 nl/min. The liquid phase separation gradient is as follows: Buffer solution A is 0.1% formic acid aqueous solution, B is 0.1% formic acid, acetonitrile and water mixture solution (acetonitrile: 95%). 0–3 min, 2–7% B; 3–48 min, 7–35% B; 48–53 min, 35–90% B; 53–60 min, 90% B. Data-dependent acquisition (DDA) mass spectrometry was performed with a Q-Exactive HFX mass spectrometer (Thermo Scientific) after peptide separation. The analysis time was 60 min, detection mode: positive ion, parent ion scanning range: 300–1800 *m*/*z*, primary mass spectrometric resolution: 70,000 *m*/*z* 200. Secondary mass spectrogram of 20 parent ions with the highest intensity was acquired after each full scan. The software MaxQuant1.6.1.0 and Uniprot Protein Database were used to analyze the mass spectrometry database.

### 4.6. Macrophage Culture and Treatment

RAW 264.7 macrophages were incubated in a DMEM/high-glucose culture medium that was mixed with 10% FBS and 1% antibiotics (100 U/mL penicillin and 100 U/mL streptomycin) and placed in a CO_2_ incubator (5% CO_2_, 37 °C). When they had grown to 80% of the area of the culture flask, the macrophages were washed down and planted in 96- or 6-well plates with cell density of 5 × 10^3^ cells/well, and incubated for 24 h. Then, different concentrations of EPS11 (0–200 μg/mL) or mixed with LPS (1.0 μg/mL) were added to the corresponding wells for 24 h. l-NMMA (50 μmol/L) or aspirin (200 μg/mL) was used as the positive drug, and untreated macrophages were used as the control group.

### 4.7. Assessment of Cell Viability

The effect of EPS11 on the viability of RAW 264.7 macrophages was evaluated by MTT assay [43]. To rule out the potential influences of drugs on cell viability, the macrophages were treated by different concentrations of EPS11 (6.25, 12.5, 25, 50, 80, 100, and 200 μg/mL) or with LPS (1.0 μg/mL) in 96-well plates for 24 h. FBS-free medium (100 μL) and 5.0 mg/mL MTT stock solution (10 μL) were immediately added to each well and incubated for 4 h to form the formazan. Then, the supernatants were carefully discarded, and DMSO (50 μL) was added to entirely dissolve the formazan. The absorbance was measured at 570 nm. Cell viability is presented in the form of percentage contrasted with the control group.

### 4.8. Active Fraction Appraisal

As EPS11 is a proteoglycan, it is not known which part performs the anti-inflammatory activity. To identify the active fraction, EPS11 (200 μg/mL) was treated with proteinase K (50 μg/mL) and NaIO4 (15 mmol/L), respectively. Then, the LPS-induced RAW264.7 macrophages were incubated with EPS11 that was treated in two different ways. The levels of NO production were detected to evaluate anti-inflammatory activity of each fraction. 

### 4.9. Determination of NO Production 

NO concentration in the supernatant was measured using the Griess reagent [44]. RAW264.7 macrophages were treated by different concentrations of EPS11 (6.25, 12.5, 25, 50, 100, and 200 μg/mL) or with LPS (1.0 μg/mL) in 96-well plates for 24 h. Cell-culture supernatants (50 μL) were sucked out into a new plate, and Griess reagents I (50 μL) and II (50 μL) were added in each well. The NO concentration was calculated according to the standard curve, which was measured at 540 nm.

### 4.10. COX-2 and Cytokine Assays

RAW264.7 macrophages were cotreated by various concentrations of EPS11 (50, 100, and 200 μg/mL) and LPS (1.0 μg/mL) in 6-well plates for 24 h. The cell-culture supernatants in each well were collected and centrifuged to measure the levels of COX-2 and inflammatory cytokines (TNF-α, IL-1β, and IL-6) using commercial ELISA kits according to operating instructions.

### 4.11. Western Blot Analysis

RAW 264.7 macrophages were cotreated by EPS11 (50, 100, and 200 μg/mL) and LPS (1.0 μg/mL) in 6-well plates for 24 h. Then, the total proteins of the macrophages were extracted by 100 μL RIPA lysis buffer (G-CLONE, Beijing, China). The protein content of every group was determined by BCA protein assay kit (Solarbio, Beijing, China). The total proteins were corrected to equivalents and denatured with loading buffer. The target proteins were separated by SDS-polyacrylamide gel electrophoresis (SDS-PAGE) and transferred to a nitrocellulose-filter (NC) membrane. Then, the membrane was sealed with 5% skim milk, hybridized with primary antibodies, and incubated with a secondary antibody in sequence. Enhanced chemiluminescence (Vazyme, Nanjing, China) was applied to detect protein blots using a ChemiDoc MP system (Bio-Rad Laboratories, Hercules, CA, USA). The relative grayscale of the protein band was calculated by using Image J v.1.52a (National Institutes of Health, Bethesda, MD, USA).

### 4.12. Statistical Analysis

The statistical analysis was performed using Prism 6 (version 6.01, GraphPad Software, San Diego, CA, USA). All data are presented as the average value ± SD from at least three independent experiments. The significance was analyzed between the two groups using an unpaired t-test. A *p* value < 0.05 was considered to be the level of statistical significance (* *p* < 0.05, ** *p* < 0.01, *** *p* < 0.001).

## 5. Conclusions

The current study suggested that marine bacterial proteoglycan (EPS11) exhibited strong anti-inflammatory activity. This could significantly decrease the generation of NO, COX-2, TNF-α, IL-1β, and IL-6 through suppressing the phosphorylation of P38, ERK1/2, and JNK in MAPK pathways, and the expression of NF-κB P65 in LPS-stimulated RAW 264.7 macrophages. Moreover, EPS11 may mainly inhibit the NF-κB pathway to execute this anti-inflammatory effect. Future studies are expected to confirm the anti-inflammatory effect in inflammatory animal models, and ultimately demonstrate the potential application of EPS11 in the treatment of inflammatory diseases.

## Figures and Tables

**Figure 1 marinedrugs-18-00585-f001:**
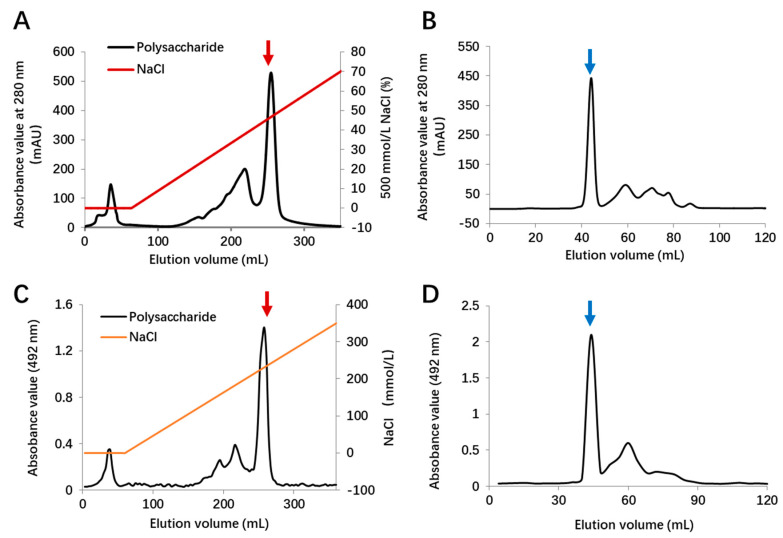
Purification of marine *Bacillus* sp. BS11 (EPS11). (**A**) Elution spectra of diethylaminoethyl cellulose (DEAE) Fast Flow ion-exchange column chromatography of protein. (**B**) Elution spectra of HiLoad^TM^ 16/600 Superdex^TM^ 200 gel-column chromatography of first purified fraction (red arrow in (**A**)). (**C**) Elution spectra of diethylaminoethyl cellulose (DEAE) Fast Flow ion-exchange column chromatography of crude polysaccharide. (**D**) Elution spectra of HiLoad^TM^ 16/600 Superdex^TM^ 200 gel-column chromatography of first purified fraction (red arrow in (**C**)). Final purified fraction (blue arrow) was collected, dialyzed, and lyophilized.

**Figure 2 marinedrugs-18-00585-f002:**
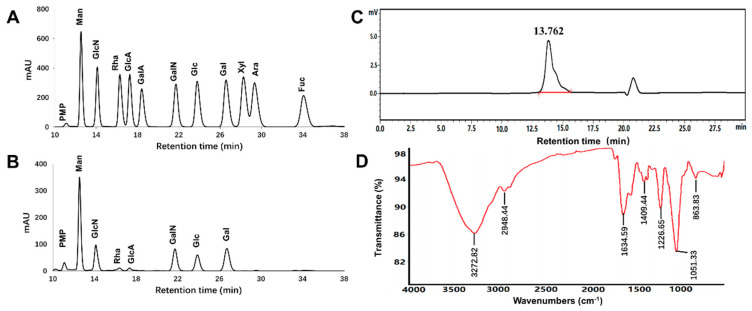
Physicochemical properties of EPS11. (**A**) Chromatography of monosaccharides with mixed standards including eleven sugars (Man, GlcN, Rha, GlcA, GalA, GalN, Glc, Gal, Xyl, Ara, Fuc). (**B**) Chromatography of EPS11 for monosaccharide-composition analysis. (**C**) Chromatography of EPS11 for molecular-weight determination. (**D**) FT–IR spectrum of EPS11.

**Figure 3 marinedrugs-18-00585-f003:**
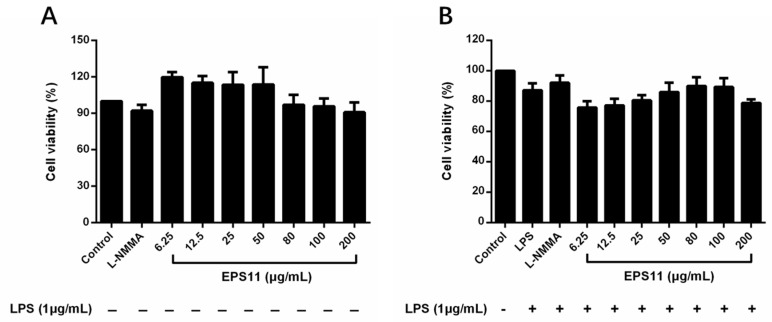
Effect of EPS11 on RAW 264.7 macrophage viability. (**A**) Macrophages cultured by EPS11 (6.25, 12.5, 25, 50, 80, 100, and 200 μg/mL) alone for 24 h. (**B**) Macrophages cotreated by EPS11 (6.25, 12.5, 25, 50, 80, 100, and 200 μg/mL) and lipopolysaccharides (LPS; 1 μg/mL) for 24 h. Data presented as means ± SD (*n* = 3) from independent experiments.

**Figure 4 marinedrugs-18-00585-f004:**
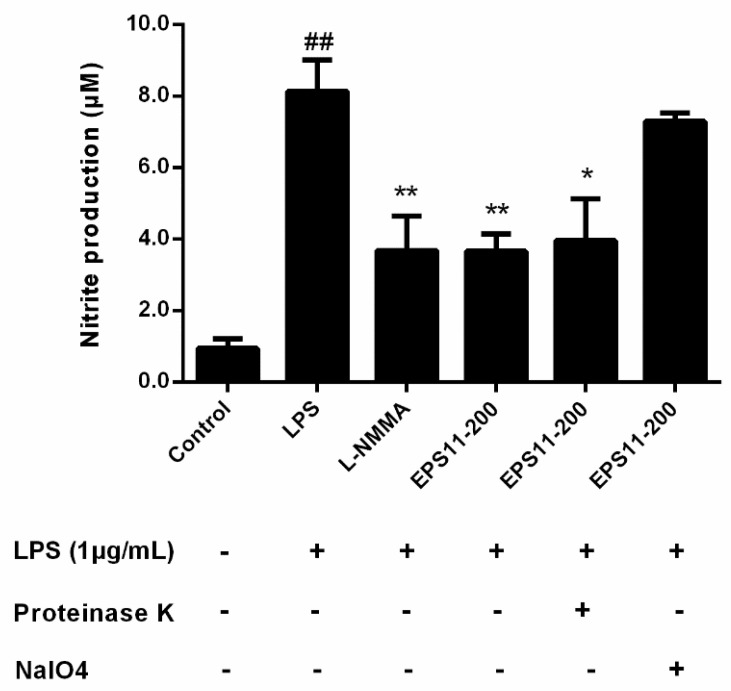
Identification of active fraction of EPS11. EPS11 (200 μg/mL) was. LPS-induced RAW264.7 macrophages were incubated with EPS11 that was treated with proteinase K (50 μg/mL) and NaIO4 (15 mmol/L), respectively. The levels of NO production were detected to evaluate the anti-inflammatory activity of each fraction. Data presented as mean ± SE (*n* = 3) from independent experiments. Significance: ^##^
*p* < 0.01 vs. control; * *p* < 0.05, ** *p* < 0.01 vs. LPS-treated.

**Figure 5 marinedrugs-18-00585-f005:**
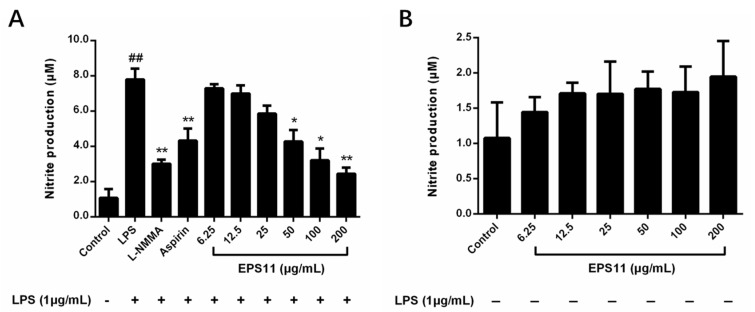
Effect of EPS11 on NO production in RAW264.7 macrophages. (**A**) Macrophages cotreated with EPS11 (6.25, 12.5, 25, 50, 100, and 200 μg/mL) and LPS (1.0 μg/mL) for 24 h. (**B**) Macrophages treated with EPS11 (6.25, 12.5, 25, 50, 100, and 200 μg/mL) alone for 24 h. Data presented as mean ± SE (n = 3) from independent experiments. Significance: ^##^
*p* < 0.01 vs. control; * *p* < 0.05, ** *p* < 0.01 vs. LPS-treated.

**Figure 6 marinedrugs-18-00585-f006:**
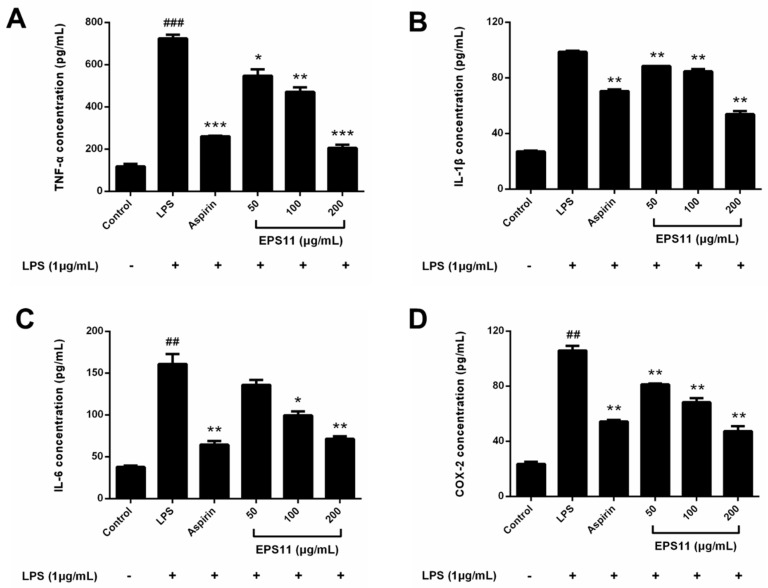
Effects of EPS11 on COX-2, TNF-α, IL-1β, and IL-6 protein expressions in LPS-stimulated RAW 264.7 macrophages. Macrophages cotreated with EPS11 (50, 100, and 200 μg/mL) and LPS (1.0 μg/mL) for 24 h. (**A**) TNF-α, (**B**) IL-1β, (**C**) IL-6, and (**D**) COX-2 determined using ELISA kits. Data presented as mean ± SE (*n* = 3) from independent experiments. Significance: ^##^
*p* < 0.01, ^###^
*p* < 0.001 vs. control; * *p* < 0.05, ** *p* < 0.01, *** *p* < 0.001 vs. LPS-treated.

**Figure 7 marinedrugs-18-00585-f007:**
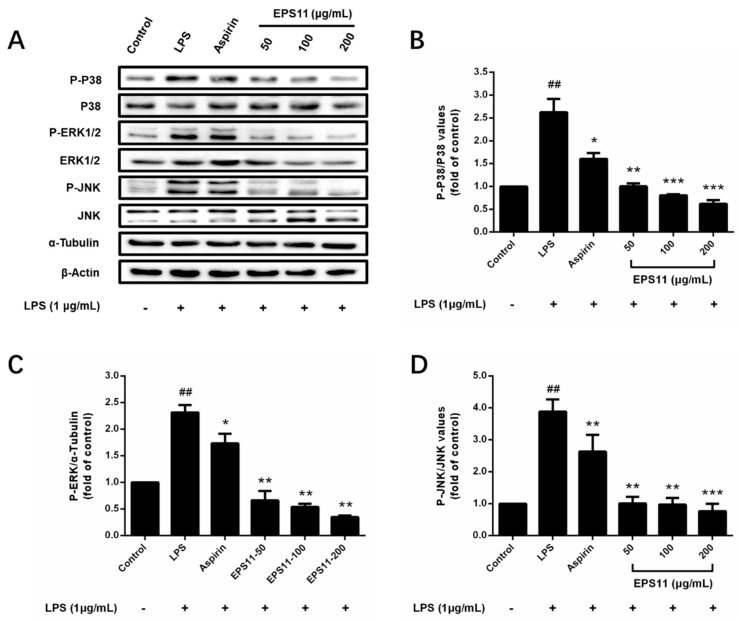
Inhibitory effects of EPS11 on mitogen-activated protein kinase (MAPK) pathway in LPS-stimulated RAW264.7 macrophages. (**A**) Protein-expression levels of P-P38, P38, P-ERK1/2, ERK1/2, P-JNK, and JNK measured by Western blot analysis. Gray value ratios of (**B**) P-P38 to P38, (**C**) P-ERK1/2 to α-tubulin, and (**D**) P-JNK to JNK. Ratio for the control was assigned a value of 1.0. Data presented as mean ± SD (*n* = 3) from independent experiments. Significance: ^##^
*p* < 0.01 vs. control; * *p* < 0.05, ** *p* < 0.01, *** *p* < 0.001 vs. LPS-treated.

**Figure 8 marinedrugs-18-00585-f008:**
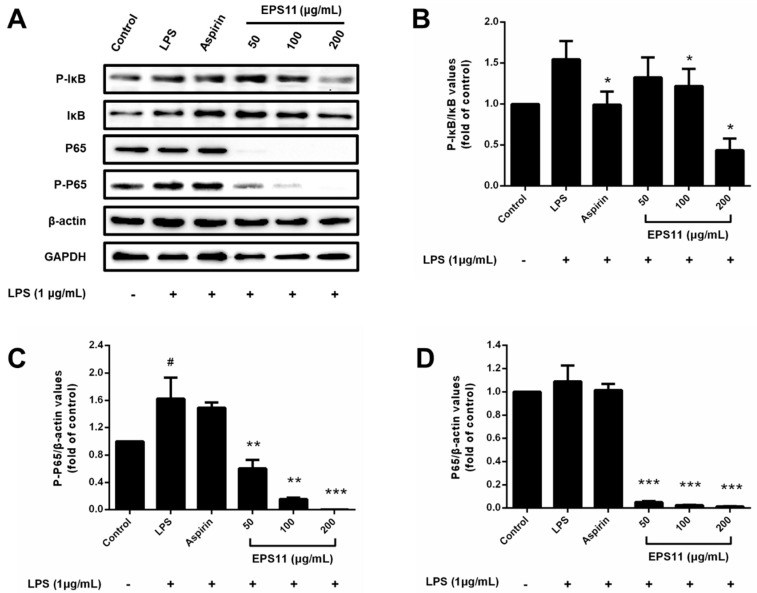
Inhibitory effect of EPS11 on NF-κB pathway in LPS-stimulated RAW264.7 macrophages. (**A**) Protein-expression levels of P-IκB, IκB, P65, and P-P65 measured by Western blot analysis. Gray value ratios of (**B**) p- IκB to IκB, (**C**) P-P65 to β-actin, and (**D**) P65 to β-actin. Ratio for control assigned a value of 1.0. Data presented as mean ± SD (*n* = 3) from independent experiments. Significance: ^#^
*p* < 0.05 vs. control; * *p* < 0.05, ** *p* < 0.01, *** *p* < 0.001 vs. LPS-treated.

**Table 1 marinedrugs-18-00585-t001:** Chemical analysis of EPS11.

EPS11	
Total sugar content	49.5%
Protein content	30.2%
Moisture content	5.6%
Average molecular weight (Da)	3.06 × 10^5^
Monosaccharide composition (molar ratio)	
Mannose (Man)	13.08
Glucosamine (GlcN)	7.84
Rhamnose (Rha)	1.00
Glucuronic acid (GlcA)	1.03
Galactosamine (GalN)	8.23
Glucose (Glc)	4.73
Galactose (Gal)	7.03

**Table 2 marinedrugs-18-00585-t002:** Amino acid composition of EPS11.

No.	Name	Retention Time (min)	Peak Area	Content (% of Total Sample)
1	Aspartic acid (Asp)	9.11	18,383.87	3.76
2	Threonine (Thr)	11.17	5331.41	0.90
3	Serine (Ser)	12.05	23,249.59	3.70
4	Glutamic acid (Glu)	13.81	35,477.82	8.46
5	Glycine (Gly)	19.95	18,134.89	2.05
6	Alanine (Ala)	21.30	12,356.06	1.65
7	Cysteic acid (Cys)	22.73	406.54	0.25
8	Valine (Val)	23.94	6759.21	1.17
9	Methionine (Met)	26.09	785.08	0.18
10	Isoleucine (Ile)	28.68	6223.54	1.23
11	Leucine (Leu)	29.88	4052.47	0.80
12	Tyrosine (Tyr)	32.22	1479.20	0.41
13	Phenylalanine (Phe)	33.38	1476.18	0.37
14	Histidine (His)	36.59	11,132.71	2.55
15	Lysine (Lys)	29.68	4326.45	0.93
16	Arginine (Arg)	44.22	1960.02	0.58
17	Proline (Pro)	15.33	1684.57	1.55
	Total		153,219.60	30.52

**Table 3 marinedrugs-18-00585-t003:** Protein analysis of EPS11.

No.	Majority Protein IDs	Protein Names	Sequence Coverage [%]	Mol. Weight [kDa]	Score
1	A0A2A9CJM4	Bacillopeptidase F	14.1	155.94	323.31
2	A0A6I2ACB9	Actin, cytoplasmic 2	30.3	41.71	119.44
3	A0A2L0R4Y2	Elongation factor Tu (Fragment)	21.4	43.05	44.24
4	A0A5A9E4B1	S8 family serine peptidase	2.8	147.43	14.79
5	A0A2S9WBY3;A0A328LLM6	F0F1 ATP synthase subunit beta (Fragment)	10.7	14.17	12.54
6	A0A2A9CFX8	Spore coat protein E	14.1	21.01	12.15
7	A0A328M8Z8	Transcription termination factor Rho	2.5	66.63	8.49
8	A0A4U1CWZ7	Unnamed (Gene names: FA727_21720)	8.9	22.85	−2.00

**Table 4 marinedrugs-18-00585-t004:** Chromatographic conditions of monosaccharide composition.

	Parameters
Mobile phase	0.1 mol/L phosphate-buffered saline (PBS; pH 6.8): acetonitrile = 83:17 (*v*/*v*, %)
Chromatographic column	ZORBAX SB-AQ C18 column (4.6 × 250 mm, 5 µm)
Temperature	30 °C
Flow rate	0.8 mL/min
Detector	Variable-wavelength detector (VWD; 245 nm)

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
