# Peer review of "Proteoglycan from Bacillus sp. BS11 Inhibits the Inflammatory Response by Suppressing the MAPK and NF-κB Pathways in Lipopolysaccharide-Induced RAW264.7 Macrophages"

_marinedrugs, 2020, doi:10.3390/md18120585_

Round 1

Reviewer 1 Report

Firstly, I would strongly urge the authors to seek the input of an English language expert to improve the language and grammar throughout the manuscript. Examples of bad sentence structure and use of language include: (line 36) Especially, inflammation not only cause pain and tissue damage, but also is the inducement of many diseases, (line 38) Therefore, a number of chemosynthesis drugs have been applied to treat those (line 42) natural polysaccharides have been given a lot of attentions attributing (line 43)Thus, discovery of novel polysaccharides with immunomodulatory function from nature is emerging as an important trend.

The major issue I have with this manuscript concerns the purification and analysis of the exopolysaccharide. Figure 1 shows ion-exchange and gel permeation chromatograms to obtain a single peak detected by UV at 280 nm. This would suggest that there is protein present, despite treatment of the EPS with Sevag’s reagent (not savage reagent as shown at line 233). The gel permeation chromatogram shown in Figure 1B suggests that the isolated peak is eluting at the exclusion limit of the column (elution at 40 mL; total bed volume 120 mL), which would suggest a molecular weight in excess of 100,000 Da. Figure 2C, molecular weight determination, shows elution at the exclusion limit of the TSK-Gel 3000PWXL column and therefore  determination of molecular weight is very difficult; the authors should show the curve obtained using dextran molecular weight standards.

Also, the monosaccharide composition is shown as a molar ratio, which only shows the proportions of the different sugars detected. The authors need to provide further data to show the amount of carbohydrate, protein (and preferably also moisture and ash) in the purified exopolysaccharide.

Given my concerns about the composition of the exopolysaccharide, what is the basis of the biological activities observed? Is it due to polysaccharide, protein, glycoprotein, or some other component of the extract?

The biological activity data appears to be conducted and reported appropriately. I have not commented further as this is not my area of expertise.

Reviewer 2 Report

The work presented by Wang and coworkers is relevant for the research community working in inflammation and in the potential therapeutic effect of natural resources. The experiments performed with the EPS11 in the macrophage cell line were relevant to determine the potential anti-inflammatory effect of the exopolysaccharide. However, the manuscript should be improved before its publication.

1) The introduction mixes some concepts of inflammation and immunity and it should be reviewed for a better understanding. In fact, it is written: “Exopolysaccharide (EPS11) was sourced from a marine Bacillus sp. BS11, which was reported to inhibit cancer cell growth via blocking macrophage adhesion in previous studies [21, 22].” However, in ref. 21 and 22 there is not any mention to macrophage adhesion, but to the regulation of some surface markers implicated in the adhesion/migration of immune and tumor cells. The text should be reviewed accordingly and the aim of the work should be more clearly stated.

2) The discussion of the results should also be improved. The authors failed to stablish more connections between the molecular mechanisms and the inflammatory molecules studied.

For instance, NFkB activation in macrophages leads to the expression of proinflammatory cytokines, such as TNFa, IL-1 and IL-6 and the differentiation into the M1 macrophage phenotype.  See ref.: Liu, T., Zhang, L., Joo, D. et al. NF-κB signaling in inflammation. Sig Transduct Target Ther 2, 17023 (2017). https://doi.org/10.1038/sigtrans.2017.23

Another example can be found in ref.:

Aktan F. iNOS-mediated nitric oxide production and its regulation. Life Sci. 2004 Jun 25;75(6):639-53. doi: 10.1016/j.lfs.2003.10.042. PMID: 15172174.

3) PGE2 and NO are described as cytokines, however they are a prostaglandin implicated in the induction of fever and a free radical that works as a signaling molecule, respectively.

4) Other minor errors in the text are listed below:

-Lines 35-36:

 However, the dysregulation of immunization may cause many pathological immune responses like inflammation, allergy and asthma

“…dysregulation of the immune system…”

-Lines 47-48:

Many external stimuli, such as LPS, carbohydrate and some cytokines can combine to the pattern recognition receptors of macrophages.

…and some cytokines can interact with /bind to the pattern recognition receptors of macrophages.

-Lines 51-54:

During inflammation, macrophages execute its function of immune response through producing various inflammatory cytokines [17, 18], such as tumor necrosis factor (TNF-α), transforming growth factor-β (TGF-β), interleukin-1 (IL-1), interleukin-6 (IL-6), prostaglandin E2 (PGE2) and nitric oxide (NO)

PGE2 and NO are not cytokines (see previous comments)

-Lines 59-61:

The in vitro immunomodulatory effect of EPS11 was investigated by determining the inflammatory mediators produced by LPS-activated RAW 264.7 macrophages.

“…was investigated by determining the inhibition of inflammatory mediators…”

-Lines 137-139:

However, the nonphosphorylation of P38 and JNK were unchanged in all groups while that of ERK1/2 decreased by EPS11. It indicated that EPS11 may act by restraining the phosphorylation of P38, ERK1/2 and JNK in MAPK pathways.

Reformulate the sentence for a better understanding.

-Figure 6: review the figure legend for the graph C (p-ERK/a-tubulin).

-Figure 7: the description of C and D was exchanged in the figure legend.

Author Response

Response to Reviewer 2 Comments

Point 1: The introduction mixes some concepts of inflammation and immunity and it should be reviewed for a better understanding. In fact, it is written: “Exopolysaccharide (EPS11) was sourced from a marine Bacillus sp. BS11, which was reported to inhibit cancer cell growth via blocking macrophage adhesion in previous studies [21, 22].” However, in ref. 21 and 22 there is not any mention to macrophage adhesion, but to the regulation of some surface markers implicated in the adhesion/migration of immune and tumor cells. The text should be reviewed accordingly and the aim of the work should be more clearly stated.

Response 1: Thank the reviewer for the comments. This is a verbal error in writing. The sentence shoud be “Exopolysaccharide (EPS11) was sourced from a marine Bacillus sp. BS11, which was reported to inhibit cancer cell growth via blocking cell adhesion in previous studies [21, 22].” We have changed the word “macrophage” to “cell” (Lines 57).

The previous studies revealed EPS11 has significant anti-tumor activities. It could inhibit cancer cell growth via blocking cell adhesion. In another anti-inflammatory assay, we found that EPS11 could inhibit the NO production in LPS stimulated RAW 264.7 macrophages. The aim of this study is to investigate its anti-inflammatory effects and preliminary mechanism.

Point 2: The discussion of the results should also be improved. The authors failed to stablish more connections between the molecular mechanisms and the inflammatory molecules studied.

For instance, NFkB activation in macrophages leads to the expression of proinflammatory cytokines, such as TNFa, IL-1 and IL-6 and the differentiation into the M1 macrophage phenotype. See ref.: Liu, T., Zhang, L., Joo, D. et al. NF-κB signaling in inflammation. Sig Transduct Target Ther 2, 17023 (2017). https://doi.org/10.1038/sigtrans.2017.23

Another example can be found in ref.:

Aktan F. iNOS-mediated nitric oxide production and its regulation. Life Sci. 2004 Jun 25;75(6):639-53. doi: 10.1016/j.lfs.2003.10.042. PMID: 15172174.

Response 2: We appreciate the reviewer for the valuable comments and these comments are very helpful to improve the quality of our paper. The work presented in these two articles provided a good reference to establish connections between the molecular mechanisms and the inflammatory molecules. The two literatures (Liu, Ting , et al. "NF-κB signaling in inflammation." Signal Transduction & Targeted Therapy, 2017, 2:17023.) and (Aktan, F. INOS-Mediated Nitric Oxide Production and Its Regulation. Life Sciences, 2004, 75: 639-653.) have been cited in the revised manuscript as suggested by the reviewer (Line 397, 425-426). And their conclusions have been added in the discussion:

“NO, as an important indicator of inflammation, was produced and regulated by nitric oxide synthase (iNOS).” (Lines 113-114);

“As we known, NO is produced and regulated by iNOS, which is activated by immunostimulating cytokines (such as IL-1β, IL-6, TNF-α) through the activation of inducible nuclear factors, including NF-κB [23]. The decrease of inflammatory mediators indicated that the activity of inflammation-related signaling pathways was inhibited.” (Lines 186-190);

Add the sentences: “These stimuli could activate ERK, JNK and P38, and then act on their respective substrates to affect the activity of a variety of transcription factors, thereby regulating the genes expression of various cytokines such as TNF, IL-1, IL-6, and IL-8. In turn, TNF-α, IL-1 and other inflammatory mediators could activate different MAPKs and regulate the production of other inflammatory mediators by promoting or inhibiting the transcription of genes.” (Lines 194-198);

Rewrite the sentences: “It indicated that EPS11 could suppress the activation of MAPK pathways and reduce the intracellular transduction of inflammatory signals, which could down-regulate the production of inflammatory mediators by inhibiting the transcription of genes. It might represent one of the possible mechanisms underlying the anti-inflammatory activity of EPS11.” (Lines 203-206);

Add the sentences: “The activation of NF-κB in macrophages can promote the expression of pro-inflammatory cytokines (such as TNF-α, IL-1 and IL-6) and the differentiation into the M1 macrophage phenotype” (Lines 213-215).

Point 3: PGE2 and NO are described as cytokines, however they are a prostaglandin implicated in the induction of fever and a free radical that works as a signaling molecule, respectively.

Response 3: Thank the reviewer for the comments. We realized that our statement was improper and confusing, and knew that cytokines are low molecular weight soluble proteins induced by immunogen, mitogen or other stimulants. We have now rewritten this sentence “During inflammation, macrophages execute its function of immune response through producing various inflammatory cytokines and mediators [17, 18], such as tumor necrosis factor (TNF-α), transforming growth factor-β (TGF-β), interleukin-1 (IL-1), interleukin-6 (IL-6), prostaglandin E2 (PGE2) and nitric oxide (NO) [19, 20]” (Lines 51).

Point 4: Other minor errors in the text are listed below:

-Lines 35-36:

However, the dysregulation of immunization may cause many pathological immune responses like inflammation, allergy and asthma

“…dysregulation of the immune system…”

-Lines 47-48:

Many external stimuli, such as LPS, carbohydrate and some cytokines can combine to the pattern recognition receptors of macrophages.

…and some cytokines can interact with /bind to the pattern recognition receptors of macrophages.

-Lines 51-54:

During inflammation, macrophages execute its function of immune response through producing various inflammatory cytokines [17, 18], such as tumor necrosis factor (TNF-α), transforming growth factor-β (TGF-β), interleukin-1 (IL-1), interleukin-6 (IL-6), prostaglandin E2 (PGE2) and nitric oxide (NO)

PGE2 and NO are not cytokines (see previous comments)

-Lines 59-61:

The in vitro immunomodulatory effect of EPS11 was investigated by determining the inflammatory mediators produced by LPS-activated RAW 264.7 macrophages.

“…was investigated by determining the inhibition of inflammatory mediators…”

-Lines 137-139:

However, the nonphosphorylation of P38 and JNK were unchanged in all groups while that of ERK1/2 decreased by EPS11. It indicated that EPS11 may act by restraining the phosphorylation of P38, ERK1/2 and JNK in MAPK pathways.

Reformulate the sentence for a better understanding.

-Figure 6: review the figure legend for the graph C (p-ERK/a-tubulin).

-Figure 7: the description of C and D was exchanged in the figure legend.

Response 4: Very thanks for the reviewer's valuable comments. We have corrected these errors as suggested by the reviewer.

“However, the dysregulation of the immune system may cause many pathological immune responses like inflammation, allergy and asthma” (Lines 35-36);

“Many external stimuli, such as LPS, carbohydrate and some cytokines can interact with the pattern recognition receptors of macrophages” (Lines 46-47);

“During inflammation, macrophages execute its function of immune response through producing various inflammatory cytokines and mediators” (Lines 50-51);

“The in vitro immunomodulatory effect of EPS11 was investigated by determining the inhibition of inflammatory mediators produced by LPS-activated RAW 264.7 macrophages.” (Lines 59-61);

Rewrite the sentences: “In the test concentration, EPS11 dramatically depressed the phosphorylated P38, JNK, ERK1/2, and nonphosphorylated ERK1/2 in a concentration-dependent manner (Fig. 6B-D). However, the nonphosphorylated P38 and JNK were unchanged in all groups. It indicated that EPS11 may act by inhibiting the phosphorylation of P38, ERK1/2 and JNK, and the expression of ERK1/2 in MAPK pathways.” (Lines 145-149);

Figure 6: the figure legend for the graph C has changed (p-ERK/α-Tubulin). (Line 153)

Figure 7: the description of C and D has exchanged. (Lines 171-172).

Reviewer 3 Report

This is article describes some interesting biological activity of a lipopolysaccharide. Unfortunately, the biological data lacks meaning without more structural detail being disclosed regarding the purity and structure of the oligosaccharide, e.g., what is the carbohydrate sequence, what is the number of repeat units, what is the avg. molecular weight and the molecular weight range, are the sugars sulfated?). Without these sorts of structural details, the article can not be published.

Round 2

Reviewer 1 Report

The authors have improved the language and grammar using a professional editing service. However, I feel that there are still grammatical errors that detract from the readability of the paper. An example of this is at line 41, where I would suggest that the sentence is changed to read, “In recent years, much attention has been paid to natural polysaccharides, due to their non-toxic properties and immunomodulatory activities.” Please read the entire manuscript and make further improvements.

The analysis of the EPS has been improved, but now raises the question of whether you can call this a polysaccharide? Should it be called a proteoglycan, in which case you need to show that the UV trace is coincidental with a refractive index trace? If the EPS is treated with protease can the protein content be reduced and is there is major reduction the molecular weight of the material? Given the high protein content, further analysis of this portion of the material should also be conducted – amino-acid analysis would be appropriate.

Reviewer 3 Report

I as a reader am not interested in knowing the biological activity a an isolate, which may or may not be one thing and I am given no structural information as to what it is other than the components that make up the MIXTURE. I can not see how this can be published without a the minimum LCMS data. The simple fact is that the authors believe they are describing the biological properties of a glycoprotein, if this is true the molecule weight of this glycoprotein needs to be reported and the percent that species is in the isolate needs to be reported. If it is a spectrum of related glycoproteins than that spectrum needs to be reported (minimally the molecular weigh range it represents).
